# Spatial and Temporal Variations in Grassland Production from 2006 to 2015 in Mongolia Along the China–Mongolia Railway

**Ge Li** [1,2], **Juanle Wang** [1,3,*], **Yanjie Wang** [1,4], **Haishuo Wei** [1,2], **Altansukh Ochir** [5], **Davaadorj Davaasuren** [6], **Sonomdagva Chonokhuu** [5] and **Elbegjargal Nasanbat** [7]

1 State Key Laboratory of Resources and Environmental Information System, Institute of Geographic Sciences and Natural Resources Research, Chinese Academy of Sciences, Beijing 100101, China; lig@lreis.ac.cn (G.L.); wangyanjie@lreis.ac.cn (Y.W.); weihs@lreis.ac.cn (H.W.)

2 School of Civil and Architectural Engineering, Shandong University of Technology, Zibo 255049, China

3 Jiangsu Center for Collaborative Innovation in Geographical Information Resource Development and Application, Nanjing 210023, China

4 College of Geoscience and Surveying Engineering, China University of Mining & Technology (Beijing), Beijing 100083, China

5 School of Engineering and Applied Sciences, National University of Mongolia, Ulaanbaatar 210646, Mongolia; altansukh@seas.num.edu.mn (A.O.); sonomdagva@seas.num.edu.mn (S.C.)

6 School of the Art & Sciences, National University of Mongolia, Ulaanbaatar 14201, Mongolia; davaadorj@num.edu.mn

7 Information and Research Institute of Meteorology, Hydrology and Environment, Ulaanbaatar 15160, Mongolia; n_elbegjargal@yahoo.com

* Correspondence: wangjl@igsnrr.ac.cn; Tel.: +86-10-6488-8016

**Abstract:** Grassland biomass is the embodiment of grassland productivity, and the material basis for the maintenance of the grassland ecosystem. Grassland is the main vegetation type in the Mongolian Plateau. Grassland changes in the core region of the China–Mongolia–Russia Economic Corridor of the Belt and Road Initiative have an important impact on regional ecology, environmental conservation, and sustainable development. This study established three types of models for estimating grassland production through statistical analysis methods using NDVI, EVI, MSAVI, and PsnNet remote sensing indices retrieved from a Moderate Resolution Imaging Spectroradiometer (MODIS) dataset. This was combined with ground-measured grassland data and meteorological data. Based on model evaluation, the spatial and temporal distribution and variation characteristics of grassland along the Mongolia part of the China–Mongolia Railway were obtained through inversion for the period from 2006 to 2015. The results showed that all the models had good simulation effects. The optimal model was an exponential model based on MSAVI—with its simulation accuracy reaching 78%. Grassland production in the study area has increased slightly in the past ten years, with little change in the first five years and a fluctuating increase in the next five years. The average grassland production (per unit production) in the past ten years was 3400.39 kg/ha and the average total production was $9707.88 \times 10^4$ t. Grassland production increased slightly in most areas along the railway, and in some areas it continued to decline. The regional spatial distribution of increased and decreased grassland production was significantly different. With better grassland resources in the northeastern part of the study area—the area around Chinggis City and the capital of Hentiy Province—had the most significant growth. However, the southern Gobi area—with its trend towards land degradation in the area where the southern Gobi and desert steppe transitions to steppe and dry steppe—had a significant decrease. This meant that the risk of grassland degradation still existed. There were also quantitative and spatial differences in the areas where grassland production decreased on both sides of the railway. The decrease in grassland production on the western side of the railway was more obvious than on the eastern side, and the reduction area was dispersed on the western side

and relatively concentrated on the eastern side. In future research, the identification of key areas of grassland degradation along the China–Mongolia Railway as well as its driving forces should be investigated further.

**Keywords:** China–Mongolia Railway; remote sensing; estimation model; grassland production; sustainable development

## 1. Introduction

The grassland ecosystem is one of the most important terrestrial ecosystems on earth. It is not only an important geographical and ecological boundary but is also the main material basis for the development of animal husbandry [1,2]. Mongolia is rich in grassland resources, with vast natural grasslands, and is one of the world's most important centers of grassland animal husbandry in the world. For a long time, grassland degradation has occurred in some areas of Mongolia due to natural factors and human activities, and this poses a serious threat to the social and economic development and regional ecological balance there. Grassland degradation has become one of the most serious environmental problems in Mongolia and is jeopardizing the sustainable development of the area [3].

The China–Mongolia–Russia Economic Corridor is one of the key corridors proposed by the Belt and Road Initiative [4]. It entered the substantive promotion stage in 2015, and is an important bridge for linking China, Mongolia, Russia, and the European economic circle. The China–Mongolia Railway is an important channel for the China–Mongolia–Russia Economic Corridor. It is currently the only railway port connecting China and Mongolia. Strengthening the monitoring of grassland resources in this area is highly important for the sustainable development of grassland ecosystems in the Belt and Road region.

Grassland biomass is an indicator of grassland productivity and an important basis for grassland resource management. The traditional grassland monitoring method is mainly based on field investigation, thus being costly, time-consuming, and inappropriate for large-scale analysis [5]. Compared with traditional methods, remote sensing methods for estimating grassland production have the advantages of time efficiency and wide coverage, providing fast and accurate grassland information for grassland resources and animal husbandry development [6]. Chen et al. [7] established a grassland biomass estimation model suitable for grassland in China, based on NOAA/AVHRR data, and estimated the grassland biomass in Changji Prefecture of Xinjiang province from 1991 to 1993. Mao et al. [8] analyzed the relationship between grassland production data and NOAA/AVHRR (Advanced Very High Resolution Radiometer) NDVI (normalized difference vegetation index) in the Qinghai Province, and obtained grassland production in 2003–2004 by establishing a grass yield estimation model. Yang et al. [9] established the estimation model of grassland production in the Qinghai Province through the 2003 and 2004 Moderate Resolution Imaging Spectroradiometer (MODIS) EVI data, and found that the exponential model was more suitable for grassland estimation in the region. Liang et al. [10] studied the grassland biomass in southern pastoral area of the Gansu Province in China from 2001 to 2008 through MODIS data, and found that the estimation model based on MODIS EVI could better estimate the grassland biomass in this region, with an average accuracy of 76.7%. Li et al. [11] established an estimation model of grassland production in the Qinghai Province based on MODIS NDVI and ground sample data in 2007. They analyzed the change in grassland production in this area. Lv et al. [12] used MODIS NPP products to estimate the grassland production information in the Three-River Source Region of the Tibetan Plateau over the past ten years and analyzed its temporal and spatial variation characteristics. Furthermore, some researchers have carried out research on biomass in arid and semi-arid areas. Zhao et al. [13] proved that NDVI had a good effect on the grassland production estimation in Xilinguole and Inner Mongolia of China. Tucker et al. [14] studied the grassland biomass of northern Senegal, he found that there was a correlation between biomass and

NOAA data, and estimated the biomass for the area from 1980 to 1984. Todd et al. [15] monitored the vegetation biomass of Eastern Colorado in 1991 through Landsat TM data, and found that red band was more sensitive to green vegetation and more suitable for biomass estimation. Fontana et al. [16] analyzed the relationship between MODIS NDVI, EVI, and crop yields in northeastern Australia, and found a good correlation between EVI and crop yields—this result can be used for biomass estimation studies in the region. Rahetlah [17] used SPOT NDVI data to establish a grassland biomass estimation model suitable for the Madagascar region, which showed that NDVI can be used for the dynamic monitoring of pasture. Li [18] used MODIS NDVI and EVI data to study the grassland biomass in northern Xinjiang from 2004 to 2006, and found that EVI is more suitable for grassland biomass estimation in this region.

In general, research on grassland production in arid and semi-arid areas is mainly concentrated in Inner Mongolia and the Qinghai provinces of China, with relatively little research on Mongolian grassland production. Most studies focus on the analysis of vegetation coverage changes in the Mongolian Plateau, or on the grassland production in a year or a few years in the same region, and there is a lack of continuous monitoring. This study is based on ground survey data of the grassland along the China–Mongolia Railway, combined with the MODIS remote-sensing image and meteorological data. The aim of this work is to define an estimation model of grassland production suitable for the high-altitude and arid environment of the region, to study the spatial and temporal distribution and change characteristics of grassland production from 2006 to 2015.

## 2. Materials and Methods

### 2.1. Study Area

The Mongolia portion of the China–Mongolia Railway is located in the central and eastern part of Mongolia and was built between 1947 and 1955. The Sukhbaatar city north of the railway is connected to Naushki, Russia; and the Zamyu Uud city in the south is bordered by Erenhot, China. With an annual cargo capacity of 6.2 million tons, it is the most important railway line connecting China, Mongolia, and Russia. This study considers the 200 km buffer zone along the Mongolian portion of the China–Mongolia Railway (Figure 1). This zone has a typical continental temperate grassland climate, with large temperature differences between day and night. Winter is long and cold; and summer is short and hot. It receives low precipitation that mainly occurs during summer [19]. The land cover types along the railway are mainly grassland and, from north to south—forest, forest grassland, typical grassland, desert grassland, bare land, and Gobi Desert. The study area spans 13 provinces of Mongolia that represent the most populated part of the state. The main economic industries are animal husbandry and mining, such as coal and iron mines.

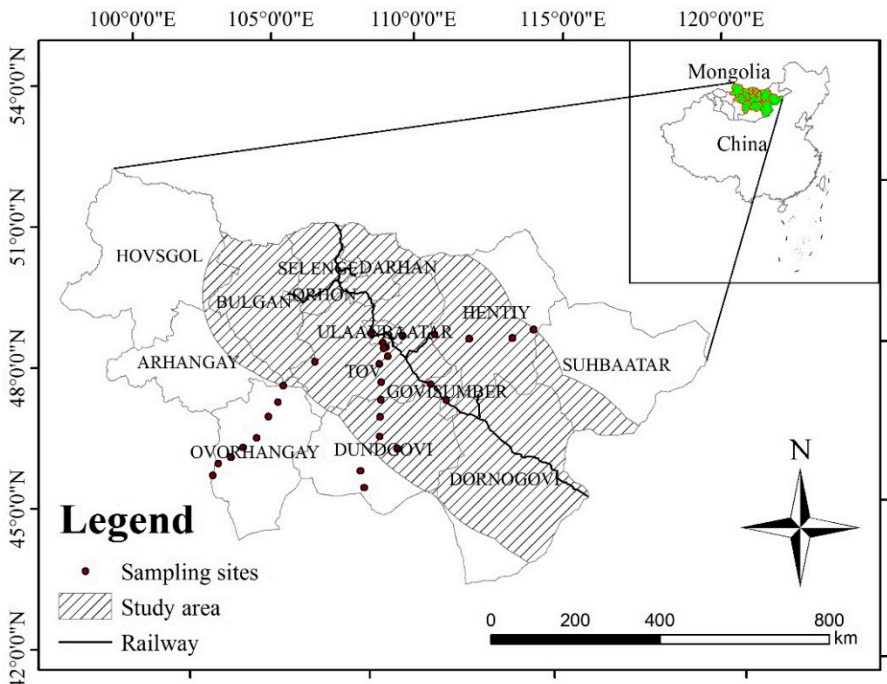

**Figure 1.** Location of study area.

*2.2. Data Sources*

(1)  Remote sensing data. The remote sensing data are the MOD13Q1 and MOD17A2H data products from the National Aeronautics and Space Administration (NASA) of the United States. MODIS data products can be divided into four topics according to data characteristics—ocean, land, atmosphere, and ice and snow. The data used in this study are MODIS land data products. The MOD13Q1 data product is called MODIS/Terra Vegetation Indices 16-Day L3 Global 250 m SIN Grid, with a time resolution of 16 day and spatial resolution of 250 m. The MOD13Q1 data has 12 bands, including two primary vegetation layers, NDVI and EVI, which can be used for vegetation condition and land cover change monitoring. The MOD17A2H data product is called MODIS/TERRA Gross Primary Productivity 8-Day L4 Global 500 m SIN Grid, with a time resolution of 8 day and spatial resolution of 500 m. The MOD17A2H has three bands, including information on Gross Primary Productivity (GPP) and net photosynthesis (PSN), which can be used to calculate terrestrial energy, carbon, water cycle processes, and biogeochemistry of vegetation. The original dataset is in HDF format and has been processed via atmospheric correction and geometric correction. The MODIS remote sensing image was set to a unified UTM projection by MODIS Reprojection Tools (MRT) software.

(2)  Ground measured data. The field survey data were collected by the research team supported by the Institute of Geography, Mongolian Academy of Sciences and Mongolia National University. The sampling time was August 2013 and August 2014, which was a period of vigorous vegetation growth. The survey included grassland production, grassland type, coordinates, terrain, and landscape description. The sample plot size was 10 m × 10 m, and three 0.5 m × 0.5 m sample squares were randomly selected. The grass in the sample square was collected by ground-level mowing, then weighed and recorded, taking the average of the three sample data as the actual grassland production data.

(3)  Other data. Socioeconomic data (population, livestock data, etc.), mean annual temperature, and mean annual precipitation data were obtained from the official website of the Mongolian National Bureau of Statistics [20] (http://www.en.nso.mn/). The meteorological data were the annual temperature and precipitation data of Mongolia from 2006 to 2015, including

22 meteorological stations. The meteorological station data were interpolated by ArcGIS 10.2 software. Mongolian administrative boundary data, including grassland type and DEM data (1 km resolution) were provided by the Thematic Database for Human-Earth System, Chinese Academy of Sciences (http://www.data.ac.cn).

*2.3. Methods*

(1) Vegetation index acquisition

According to the characteristics of the study area and referring to relevant research, four indices—NDVI, EVI, MSAVI, and PsnNet—were selected to estimate the grassland production. NDVI, EVI, and PsnNet were obtained from NASA. MSAVI was calculated by research team based on the red and near-infrared bands according to the formula. The formula for each index is as follows:

$$\text{EVI} = 2.5 \times \frac{\rho_{\text{NIR}} - \rho_{\text{RED}}}{\rho_{\text{NIR}} + 6.0\rho_{\text{RED}} - 7.5\rho_{\text{BLUE}} + 1}; \tag{1}$$

$$\text{NDVI} = \frac{\rho_{\text{NIR}} - \rho_{\text{RED}}}{\rho_{\text{NIR}} + \rho_{\text{RED}}}; \tag{2}$$

$$\text{MSAVI} = \frac{2\rho_{\text{NIR}} + 1 - \sqrt{(2\rho_{\text{NIR}} + 1)^2 - 8(\rho_{\text{NIR}} - \rho_{\text{RED}})}}{2}; \tag{3}$$

$$\text{PsnNet} = \text{GPP} - \text{Leaf\_MR} - \text{Froot\_MR}; \tag{4}$$

where $\rho_{\text{NIR}}$ is the near-infrared radiation reflectance, $\rho_{\text{RED}}$ is the red band reflectance, $\rho_{\text{BLUE}}$ is the blue band reflectance, GPP is the Gross Primary Productivity, Leaf_MR is the leaf autotrophic respiration consumption, and Froot_MR is the root autotrophic respiration consumption.

(2) Model building and accuracy verification

A correlation analysis was performed between grassland production and each vegetation index, as well as mean annual temperature, mean annual precipitation, and DEM data. On this basis, the factors with higher correlation with grassland production were selected for modeling. Ground-measured data was collected in 2013 and 2014. There are 29 samples for grassland production estimation, with 80% of the data used for modeling and 20% randomly selected for model validation. Based on the data of 22 meteorological stations in Mongolia, the mean annual temperature and precipitation data of the study area were calculated by the inverse distance weighted interpolation method. Remote sensing image data, meteorological data, and DEM data were overlay processed by ArcGIS software. We used Geographic Information System (GIS) methodology to extract the values of remote sensing data, mean annual temperature, and mean annual precipitation and DEM data for the year, corresponding to the grassland production data. We used SPSS 22.0 statistical software to carry out linear regression analysis, exponential regression analysis, and multiple linear regression analysis considering the influence of meteorological factors and constructed three types of grassland production estimation models.

The accuracy of the estimation model was evaluated by the reserved sample data. The main evaluation indicators included the average relative error and the root mean square error. The calculation formula is as follows:

$$\text{REE} = \frac{\sqrt{\frac{\sum\limits_{i=1}^{N}(Y_i - Y_i\prime)^2}{N}}}{\overline{Y_i}} \times 100\%; \tag{5}$$

$$\text{RMSE} = \sqrt{\frac{\sum\limits_{i=1}^{N}(Y_i - Y_i\prime)^2}{N}} \times 100\%; \tag{6}$$

where REE is the average relative error, RMSE is the root mean square error, N is the number of samples, $Y_i$ is the measured grassland production ($g/m^2$), $Y_i\prime$ is the estimated grassland production ($g/m^2$), and $\overline{Y}_i$ is the average measured grassland production ($g/m^2$).

(3) Interannual variation in grassland production

The one-dimensional linear regression analysis method was used to calculate the raster data of grassland production from 2006 to 2015, pixel by pixel. The change trend of grassland production in the study area in the ten years was obtained. The calculation formula is as follows [21]:

$$\beta = \frac{n \times \sum\limits_{i=1}^{n} (i \times C_i) - \sum\limits_{i=1}^{n} i \sum\limits_{i=1}^{n} C_i}{(n \times \sum\limits_{i=1}^{n} i^2) - (\sum\limits_{i=1}^{n} i)^2}, \tag{7}$$

where $\beta$ is the change rate of grassland production, *n* is the number of years (n = 10), and $C_i$ is the grassland production of year i. A positive $\beta$ means the increase of grassland production; otherwise, it means the decrease.

## 3. Results

### 3.1. Estimation Model for Grassland Production

The correlation analysis results showed that the grassland production was significantly correlated with the four remote sensing indices and the correlation coefficient was above 0.75, followed by mean annual precipitation and mean annual temperature, with correlation coefficients of 0.645 and −0.477, respectively (Table 1). The correlation between mean annual temperature and grassland production was significant at the confidence level of 0.05, and the correlation was relatively weak. There were few significant correlations between DEM data and grassland production.

**Table 1.** Correlation analysis between grassland production and impact factors.

| Element | Precipitation | Temperature | DEM | EVI | MSAVI | NDVI | PsnNet |
|---|---|---|---|---|---|---|---|
| Correlation coefficient | 0.645 ** | −0.477 * | 0.108 | 0.762 ** | 0.804 ** | 0.798 ** | 0.856 ** |
| *p* value | 0.001 | 0.022 | 0.625 | 0.000 | 0.000 | 0.000 | 0.000 |

** When the confidence is 0.01, the correlation is significant. * When the confidence is 0.05, the correlation is significant.

According to the correlation analysis results documented in Table 1, compared with other factors, the correlation between mean average temperature and grassland production was lower, and there was no significant correlation between DEM and grassland production in the Mongolian plateau where altitude was not very different. We selected mean precipitation data and four types of remote sensing indices that were significantly correlated with grassland production to establish estimation models (Table 2). The regression analysis was performed on the above five independent variables and the grassland production data by SPSS statistical software. The results showed that all the models passed the significance test ($p < 0.01$), and the determination coefficient ($R^2$) of most models was above 0.5. The overall estimation accuracy of the model based on MSAVI and NDVI was better than other models. Although NDVI usually performs well on vegetation change studies in areas with good vegetation coverage, MSAVI is extremely sensitive to changes in soil background. This study area is located in the transition area of vegetation change in Mongolia, which is an arid and semi-arid environment. The comparison showed that the exponential model based on MSAVI had the best fitting effect. The p value of the model was 0, and the determination coefficient was the highest at 0.72, which can explain 72% of the variation of the dependent variable. The model fitting accuracy results showed that the exponential model based on MSAVI had the highest accuracy at 78%, and the RMSE was 279.09 kg/ha.

Although there are still some potential uncertainties in the best-fit model, this accuracy can generally reflect the state of grassland changes in central Mongolia.

**Table 2.** Grassland production estimation model based on remote sensing.

| Parameters | Model Types | The Inversion Model | $R^2$ | Sig. | RMSE (kg/ha) | Accuracy (%) |
|---|---|---|---|---|---|---|
| EVI | Linear model | $Y = -19.490 + 384.791 * X_1$ | 0.46 | 0.000 | 423.55 | 66 |
| | Exponential model | $Y = 4.257 * \exp(12.647 * X_1)$ | 0.63 | 0.000 | 466.53 | 63 |
| | Multivariate model | $Y = -16.655 + 436.870X_1 - 0.049X_2$ | 0.47 | 0.002 | 423.91 | 66 |
| MSAVI | Linear model | $Y = -27.370 + 165.323 * X_2$ | 0.57 | 0.000 | 370.00 | 71 |
| | Exponential model | $Y = 3.594 * \exp(5.224 * X_2)$ | 0.72 | 0.000 | 279.09 | 78 |
| | Multivariate model | $Y = -18.070 + 239.539X_1 - 0.177X_2$ | 0.61 | 0.000 | 383.40 | 70 |
| NDVI | Linear model | $Y = -13.940 + 204.158 * X_3$ | 0.56 | 0.000 | 335.37 | 73 |
| | Exponential model | $Y = 5.728 * \exp(6.300 * X_3)$ | 0.68 | 0.000 | 306.68 | 76 |
| | Multivariate model | $Y = -3.192 + 264.166X_1 - 0.119X_2$ | 0.59 | 0.000 | 308.76 | 75 |
| PsnNet | Linear model | $Y = 0.482 + 0.403 * X_4$ | 0.68 | 0.000 | 389.21 | 69 |
| | Exponential model | $Y = 12.701 * \exp(0.010 * X_4)$ | 0.66 | 0.000 | 322.82 | 74 |
| | Multivariate model | $Y = 16.944 + 0.481X_1 - 0.106X_2$ | 0.70 | 0.000 | 375.42 | 70 |

*3.2. Spatial Distribution*

The spatial and temporal distribution of grassland production along the China–Mongolia Railway in Mongolia from 2006 to 2015 was obtained by inversion of the optimal exponential model based on MSAVI, $Y = 3.594 * \exp(5.224 * X)$ (Figure 2).

From 2006 to 2015, the spatial distribution of grassland production showed an increasing trend from the southeast to the northwest along the main railway line. The grassland production of the Dornogovi and Dundgovi provinces in the south was significantly lower than that of the Tov, Hentiy, and Bulgan provinces in the north. According to the geographic location of the study area described in Figure 1 and the spatial distribution results of grassland production described in Figure 2, three different levels of grassland production can be recognized in the study area. The area along the Zamyu Uud–Sainshand–Choir Railway were mainly orange–red patches, indicating that the area had the lowest grassland production, which was obviously lower than other areas along the railway. The area along the Choir–Ulaanbaatar Railway were mainly light green patches, indicating that grassland production in this area was at a medium level. The area along the Ulaanbaatar–Sukhbataar Railway were mainly dark green patches, indicating that grassland production in this area was the highest.

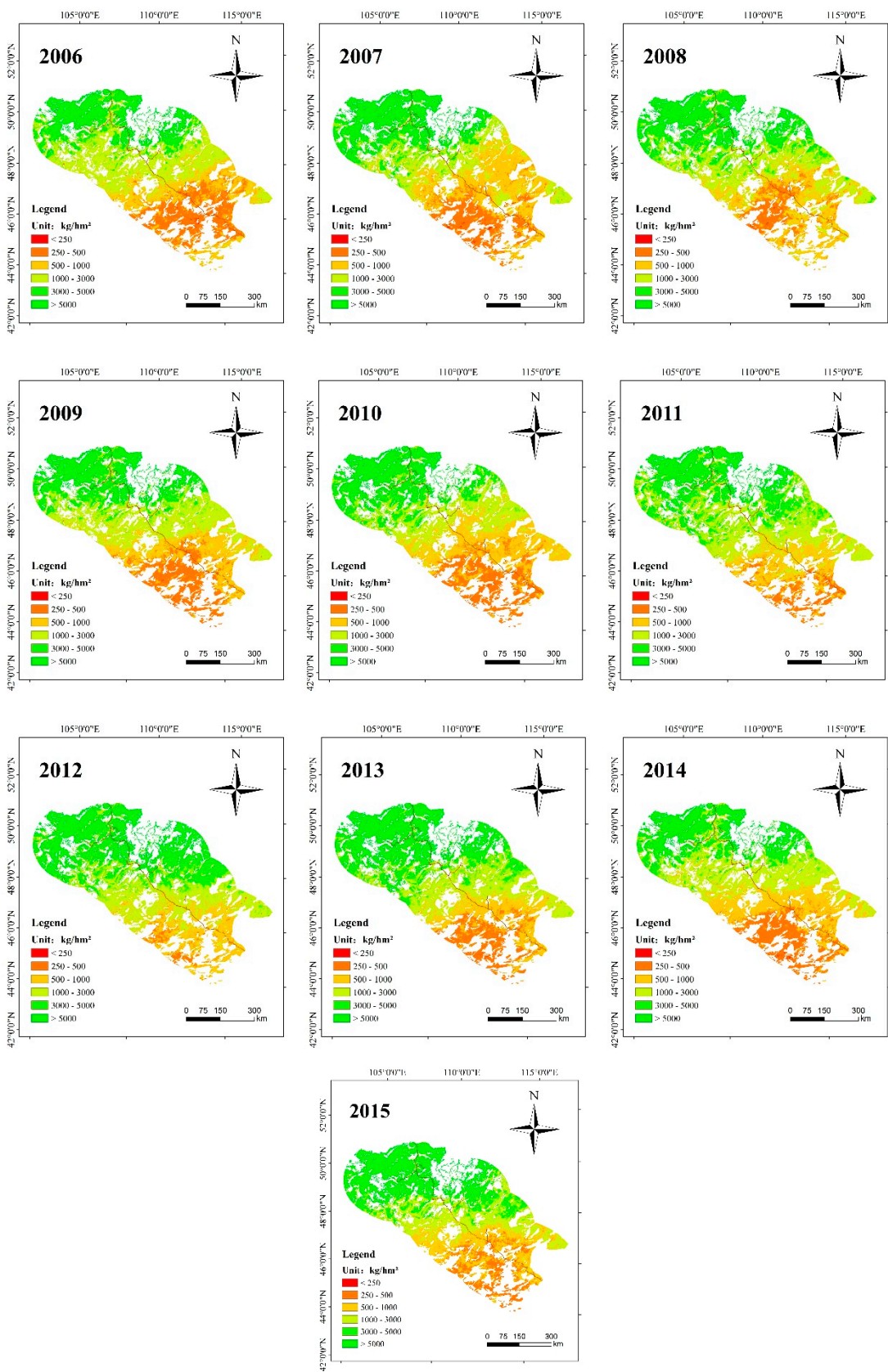

**Figure 2.** The spatial distribution of grassland production along the China–Mongolia Railway in Mongolia from 2006 to 2015.

*3.3. Temporal Distribution*

The temporal distribution characteristics of grassland production along the China–Mongolia Railway in Mongolia are shown in Figure 3. In the past ten years, grassland production has shown a slightly increasing trend. Fluctuation in the first five years was minor and grassland production did not change greatly. However, fluctuation in the latter five years showed that grassland production increased. Grassland production in 2006 was the lowest in 10 years at $8250.80 \times 10^4$ t, and the highest in 2012, reaching $11660.09 \times 10^4$ t. Average grassland production of the study area for 10 years was $9709.88 \times 10^4$ t, and average grassland production in the four years was above the average level in 2011, 2012, 2013, and 2015. The average grassland production in other years was approximately $8500 \times 10^4$ t (Table 3).

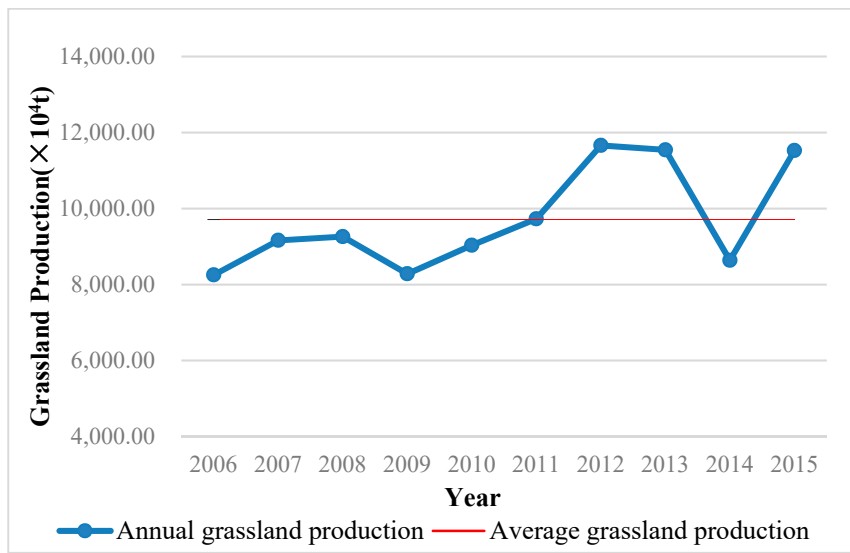

**Figure 3.** Annual grassland production along the China–Mongolia Railway in Mongolia from 2006 to 2015.

The statistical results of grassland production (total production) of grassland types are shown in Table 4. Due to the difference in grassland areas, the average grassland production of different grasslands differed greatly over ten years in high to low mountain forest steppe, steppe and dry steppe, desert steppe, mountain steppe, and mountain desert steppe. The mountain forest steppe had the highest average grassland production at $5992.43 \times 10^4$ t and was mainly distributed along the Choir–Sukhbataar Railway, while the mountain desert steppe had the lowest average grassland production at $138.76 \times 10^4$ t and was scattered along the railway. The average grassland production of steppe and dry steppe was $1536.07 \times 10^4$ t and was mainly distributed along the Choir–Zonghara Railway; desert steppe was $1533.85 \times 10^4$ t and was mainly distributed along the Zamyu Uud–Choir Railway; and mountain steppe was $506.77 \times 10^4$ t and was mainly distributed along the Sainshand–Ulaanbaatar Railway.

**Table 3.** Statistics of annual grassland production along the China–Mongolia Railway in Mongolia from 2006 to 2015.

| Yield \ Year | 2006 | 2007 | 2008 | 2009 | 2010 | 2011 | 2012 | 2013 | 2014 | 2015 | Average |
|---|---|---|---|---|---|---|---|---|---|---|---|
| Unit yield (kg/ha) | 2889.99 | 3208.89 | 3243.41 | 2900.46 | 3164.26 | 3407.40 | 4084.19 | 4043.16 | 3025.39 | 4036.78 | 3400.39 |
| Total yield ($10^4$ t) | 8250.80 | 9160.64 | 9259.79 | 8280.70 | 9033.97 | 9727.89 | 11,660.09 | 11,542.76 | 8637.24 | 11,524.93 | 9707.88 |

**Table 4.** Annual grassland production of different grassland types along the China–Mongolia Railway in Mongolia from 2006 to 2015.

| Grassland Types | Production (Total Yield) ($10^4$ t) | | | | | | | | | | |
|---|---|---|---|---|---|---|---|---|---|---|---|
| | 2006 | 2007 | 2008 | 2009 | 2010 | 2011 | 2012 | 2013 | 2014 | 2015 | Average |
| Steppe and dry steppe | 1248.65 | 1336.21 | 1504.31 | 1286.44 | 1312.83 | 1671.88 | 1999.06 | 1901.42 | 1265.87 | 1834.07 | 1536.07 |
| Mountain steppe | 385.45 | 424.22 | 473.79 | 409.83 | 434.35 | 615.38 | 702.50 | 652.40 | 394.04 | 575.78 | 506.77 |
| Mountain forest steppe | 5235.22 | 5819.43 | 5600.82 | 5173.74 | 5760.14 | 5689.95 | 6890.72 | 7041.84 | 5542.36 | 7170.06 | 5992.43 |
| Mountain desert steppe | 122.62 | 126.67 | 132.21 | 123.57 | 130.81 | 134.61 | 170.58 | 166.57 | 117.27 | 162.63 | 138.76 |
| Desert steppe | 1258.85 | 1454.11 | 1548.66 | 1287.12 | 1395.83 | 1616.06 | 1897.24 | 1780.52 | 1317.69 | 1782.39 | 1533.85 |
| Total | 8250.80 | 9160.64 | 9259.79 | 8280.70 | 9033.97 | 9727.89 | 11,660.1 | 11,542.8 | 8637.24 | 11,524.9 | 9707.88 |

Note: There are some potential errors in the grassland production estimation results in Tables 3 and 4. The $R^2$ of the optimal model is 0.72, and the regression relationship can explain 72% of the variation of the dependent variable.

### 3.4. Interannual Variation Characteristics of Grassland Production

Based on the grassland production distribution data, the change trend of grassland production in the Mongolia portion of the China–Mongolia Railway from 2006 to 2015 was obtained using a one-dimensional linear regression method (Figure 4).

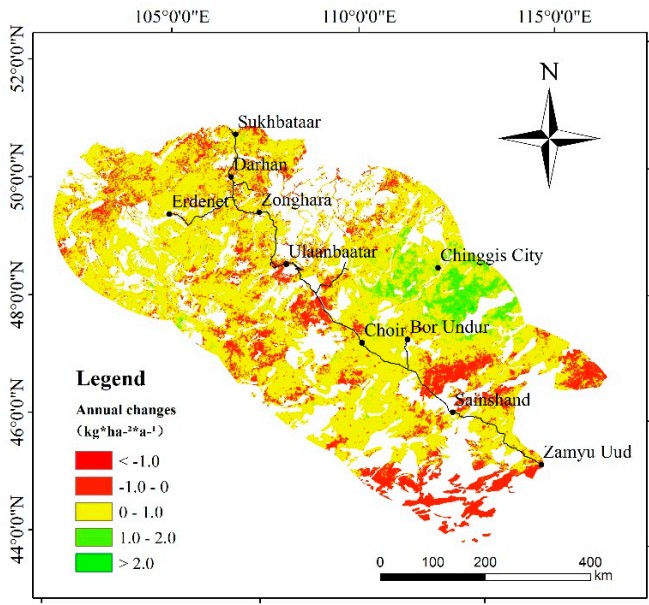

**Figure 4.** Interannual variation in grassland production along the China–Mongolia Railway in Mongolia from 2006–2015.

According to the analysis, from 2006 to 2015 the change rate of grassland production in most areas along the railway was between 0 kg·ha$^{-2}$·a$^{-1}$ and 1 kg·ha$^{-2}$·a$^{-1}$, and overall grassland production increased slightly. There were significant green patches around Chinggis City, which has traditionally had abundant grassland resources, indicating that the growth of grassland in this region is more significant, with an annual growth rates above 1.0 kg·ha$^{-2}$·a$^{-1}$ and 2.0 kg·ha$^{-2}$·a$^{-1}$. The area where the change rate of grassland production decreased showed a patchy distribution, with the change value between −1 kg·ha$^{-2}$·a$^{-1}$ and 0 kg·ha$^{-2}$·a$^{-1}$. This change was mainly in the south of the Dornogovi Province, the southwest of the Sukhbataar Province, and the southeast of the Tov Province, where the trend of land degradation was apparent. In the southern portion of the study area, grassland production in the bare land and Gobi to the grassland declined significantly. With the China–Mongolia Railway as the dividing line, the orange patches in the west side were widely distributed, and the change rate of grassland production in this area was more apparent than in the east. The area where the grassland production had declined on the west side of the railway was scattered, whereas in the east side it was densely distributed. The Sainshand–Bor Undur Railway, Choir–Ulaanbaatar Railway, and Zonghara–Sukhbataar Railway had significantly declined.

## 4. Discussion

Vegetation index is the simplest and most effective measure of vegetation cover and growth. At present, hundreds of vegetation indices have been developed (https://www.indexdatabase.de/db/i.php). The combination of vegetation index and ground-measured data has been widely used in grassland production estimation research [22]. Due to differences in topography and elevation between different study areas, the vegetation index is not universally applicable. In this study, four indices—NDVI, EVI, MSAVI, and PsnNet—were selected for grassland research. Multivariate regression analysis was conducted to consider the influence of precipitation factors and establish grassland production estimation models. Compared with other models, the overall estimation accuracy

of the models based on MSAVI and NDVI was better than that of other models, with average accuracies of 73% and 74.67%, respectively. The average estimation accuracy based on PsnNet and EVI was 71% and 65%, respectively. The optimal model was an exponential model based on MSAVI with a significance level of 0, a coefficient of determination of 0.72, and an accuracy of 78%. MSAVI has the feature of being suitable for different vegetation coverage, and can eliminate or reduce the noise of soil background [23]. It is scientific and reasonable to apply it to the grassland production estimation in this study area.

The results of this study are consistent with the changing trend of vegetation coverage in Mongolia or the Mongolian Plateau, as demonstrated by other research. For example, Batunacun et al. [24] studied the spatial distribution pattern of grassland plants in Mongolia, and found that the grassland plant species along Ulaanbaatar to Tov and Dundgovi decreased gradually; and Dai et al. [25] studied the spatial distribution of vegetation cover on the Mongolian Plateau from 2000 to 2012. Their findings are consistent with the distribution of grassland production found in this study. Zhang et al. [26] found that the vegetation coverage of the Hentiy Mountain region and the eastern part of Hangay Mountain in the Mongolian Plateau increased significantly from 2000 to 2013, while Bao et al. [27] found that the vegetation coverage of Hangay Mountain in Mongolia and its adjacent areas that extend southeast to the Dornogovi region increased significantly from 2001 to 2010. This is consistent with the change in trend of grassland production in corresponding areas during the same period found in this study.

From 2006 to 2015, grassland production along the Mongolian portion of the China–Mongolia Railway increased slightly, from 2889.99 kg/ha in 2006 to 4036.78 kg/ha in 2015. The regional characteristics of negative growth of grassland production were apparent and mainly distributed in the southeast of the Tov Province, southwest of the Sukhbataar Province and south of the Dornogovi Province. Grassland production in the Gobi area in the southern portion of the study area and the transitional areas from desert steppe to steppe and dry steppe decreased significantly. The expansion of grassland degradation transitional areas reflected that the risk of grassland degradation still exists.

The areas with decreased grassland production along the railway were mainly in the Sainshand–Bor Undur Railway, the Choir–Ulaanbaatar Railway, and the Zonghara–Sukhbataar Railway. Nowadays, research indicates that climate change, grazing activities, and rat infestations are the main causes of grassland degradation [28,29]. The study area is located in arid and semi-arid areas, with uneven distribution of precipitation and significant zonality. These factors, combined with fragile ecological conditions, make the area sensitive to climate change. Furthermore, human activities such as unsustainable land use and overgrazing also play an important role in reducing grassland production. Field investigations revealed that rat infestation is severe in some areas of the study area, and this poses problems for the development of grassland resources and animal husbandry.

From the distribution of grassland production on both sides of the China–Mongolia Railway, the reduced area of grassland production on the west side of the railway was relatively dispersed, while decreased grassland production was more significant than on the east side, and the reduced grassland production area on the east side was relatively concentrated. From the results of the correlation analysis described in Table 1, precipitation is the main climatic factor affecting grassland production. Compared with the western side, the eastern side of the railway was further from the inland area of the Mongolian Plateau. The eastern Pacific monsoon has relatively high humidity and precipitation, therefore the growth trend of grassland production in this area was more apparent. The western side of the railway has the main human population concentration in the study area. Human activities are frequent, which would directly interfere with grassland growth, and when combined with climate change impacts would exacerbate grassland degradation in this region. In the future, identifying key areas of grassland degradation along the China–Mongolia Railway needs to be further determined to provide scientific support for the protection and restoration of grassland resources.



## 5. Conclusions

This study used statistical regression methods combined with ground observation data and MODIS remote sensing data of the same period to monitor grassland production change in Mongolia along the China–Mongolia Railway, and to reflect continuous change and differences in the spatial and temporal distribution of grassland production. The results showed that with a simulation accuracy of 78%, the exponential model based on MSAVI effectively estimated grassland production. The determination coefficient of the optimal model was 0.72, so there still existed some potential uncertainty in the estimation results. The current results of grassland production from 2006 to 2015 revealed that the spatial distribution of grassland production showed an increasing trend from southeast to northwest. Grassland production along the Zamyu Uud–Sainshand–Choir Railway in the south was the lowest, while the Ulaanbaatar–Sukhbataar Railway in the north had the highest production. Grassland production changed little in the first five years and fluctuated markedly in the latter five years, showing a slight growth trend overall. Grassland production in the surrounding area of Chinggis City—with better land resources—had increased significantly. The main areas of grassland production decline were concentrated in the surrounding regions of Ulaanbaatar, Darhan, and Sukhbataar, as well as the Omnogovi Province where a land degradation trend was apparent. Our investigation of grassland production along the China–Mongolia Railway suggests that in order to curb grassland degradation and promote the green development of this region, efforts to identify key areas of grassland degradation along the China–Mongolia Railway and its driving forces should be intensified. Meanwhile, we will continue to collect sampling data and pursue better models for grassland production through a longer time-series study.

**Author Contributions:** J.W. was responsible for the research design and analysis. G.L. drafted the manuscript and was responsible for the data preparation, experiment, and analysis. Y.W. and H.W. were responsible for the data processing and archiving. A.O., D.D., S.C., and E.N. were responsible for results verification and article review.

**Funding:** This study was funded by the Strategic Priority Research Program (Class A) of the Chinese Academy of Sciences (XDA2003020302); the Asia Research Center, Mongolia and Korea Foundation for Advanced Studies, Korea (P2018-3606); and the Construction Project of the China Knowledge Center for Engineering Sciences and Technology (CKCEST-2018-2-8).

**Conflicts of Interest:** The authors declare no conflict of interest.

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
