# Peer review of "Spatial and Temporal Variations in Grassland Production from 2006 to 2015 in Mongolia Along the China–Mongolia Railway"

_sustainability, doi:10.3390/su11072177_

Round 1
Reviewer 1 Report
Title: Spatial and Temporal Variations in Grassland Production from 2006 to 2015 in Mongolia Along the China-Mongolia Railway
Overview:
The research aimed to estimate grassland biomass in Mongolia using MODIS data from 2006 to 2015, but the methods are not clearly explained and the results are overstated given the low accuracy numbers being reported.
Line 19 ‘Grassland production’ and ‘grassland productivity’ mean the same thing to me so seem redundant in this sentence.
Line 34 use hectares (ha) instead of square hectometers
Line 69 ‘Grassland production’ and ‘grassland productivity’ mean the same thing to me so seem redundant in this sentence.
The literature review seems quite brief - are there no more sources available?
Line 84 Source mentioned crop yields- is this a good match for grassland biomass?
Line 109 Does not make sense to say ‘long cold days in winter, and short hot days in summer’ daylight hours are longer in summer and shorter in winter for these latitudes.
Figure 1 needs a context map and more detailed caption.
Line 118 Define the MODIS data products and provide a reference
Line 122 What are the ‘MRT’ and why needed for this study?
Line 129 How many sample plots? Where were they located?
Lines 145-148 Were EVI, NDVI, etc. products obtained from NASA? Or authors calculated their own from the data. Not clear.
Lines 149-150 Use correct symbols for reflectance (ρ) instead of ‘PNIR’, ‘PRED’, ‘PBLUE’
Line 154 Were there 29 field plots? Sites? Sentence is not clear.
Lines 168-169 Use correct superscript for ‘m2’
Line 175 How was this equation derived? Why use MSAVI only?
Lines 182-183 These are methods, not results.
Table 2 How was accuracy calculated? Is it really model fit?
Lines 214-224 Refer to Figure 1 since readers might not be familiar with geographic locations in the study area.
Line 224 What is meant by ‘plaque’?
Figure 3 needs a legend (what is the red line?) and detailed caption. If authors want to describe a trend, add a trend line to the chart.
Tables 3 and 4 should acknowledge potentially large error in biomass estimates since best fit model has R-square = 0.72 (and accuracy 78%) that leaves 20%+ uncertainty in the results. Should be clearly spelled out.
Figure 4 Legend is not clearly labeled. What is ‘coefficient’?
Line 262 What units are ‘0 and 1’?
Line 263 Instead of ‘green patches’ use what the green patches mean (more biomass increase?)
Line 276 Remove ‘s’ from ‘Discussions’
Line 278 There are many more than 40 SVIs: https://www.indexdatabase.de/db/i.php
Line 325 What is meant by ‘obvious’? Significant?
Line 342 Acknowledge potential error in the results. A lot of uncertainty. Also not clear how well model trained on one year of data really predicts other years.
Author Response
Comments and Suggestions for Authors
Title: Spatial and Temporal Variations in Grassland Production from 2006 to 2015 in Mongolia Along the China-Mongolia Railway
Overview:
The research aimed to estimate grassland biomass in Mongolia using MODIS data from 2006 to 2015, but the methods are not clearly explained and the results are overstated given the low accuracy numbers being reported.
- Line 19 ‘Grassland production’ and ‘grassland productivity’ mean the same thing to me so seem redundant in this sentence.
Answer: Thanks. ‘Grassland production’ and ‘grassland productivity’ have different meanings. ‘Grassland production’ mean grassland biomass, ‘grassland productivity’ means grassland production capacity. In order to make them clear, ‘Grassland production’ has been replaced by ‘Grassland biomass’. See line19.
- Line 34 use hectares (ha) instead of square hectometers
Answer: Thanks for comments. The unit has been changed to ha. See line34.
- Line 69 ‘Grassland production’ and ‘grassland productivity’ mean the same thing to me so seem redundant in this sentence.
Answer: ‘Grassland production’ has been replaced by ‘Grassland biomass’. See line 69.
- The literature review seems quite brief - are there no more sources available?
Answer: Thanks for comments. We have added more references on grassland biomass monitoring in Mongolian Plateau in this section. See line 79-81 and line 87-89.
- Line 84 Source mentioned crop yields- is this a good match for grassland biomass?
Answer: Crop yield estimation is not directly related to grassland production estimation, but there are commonalities between the estimation methods. Both of these studies are based on ground-measured data and remote sensing data, and the estimation model is established by the statistical regression method for biomass research. So, in this study, some literature about crop yield estimation method is reviewed as well.
- Line 109 Does not make sense to say ‘long cold days in winter, and short hot days in summer’ daylight hours are longer in summer and shorter in winter for these latitudes.
Answer: Thanks for comments. This sentence has been revised. See line 116-117.
- Figure 1 needs a context map and more detailed caption.
Answer: Thanks for comment. Figure 1 has been updated. See line 125.
- Line 118 Define the MODIS data products and provide a reference
Answer: Thanks for comment. We have added a reference for MODIS data products. See line130-132.
- Line 122 What are the ‘MRT’ and why needed for this study?
Answer: MRT is a remote sensing image processing software, which is used for remote sensing data preprocessing in this study. This section has been revised. See line136-137.
- Line 129 How many sample plots? Where were they located?
Answer: Thanks. The sample plots information has been updated. See line173-175. There are 29 samples for modeling and validation, as shown in Figure 1. See line 125.
- Lines 145-148 Were EVI, NDVI, etc. products obtained from NASA? Or authors calculated their own from the data. Not clear.
Answer: This section has been clearly stated. See line157-158.
- Lines 149-150 Use correct symbols for reflectance (ρ) instead of ‘PNIR’, ‘PRED’, ‘PBLUE’
Answer: Thanks. This section has been revised. See line164-165.
- Line 154 Were there 29 field plots? Sites? Sentence is not clear.
Answer: The sample plots information has been updated. See line 173-175. Sample position is shown in Figure 1. See line 125.
- Lines 168-169 Use correct superscript for ‘m2’
Answer: Thanks. The unit has been revised. See line 189-190.
- Line 175 How was this equation derived? Why use MSAVI only?
Answer: This equation was introduced by reviewing the study of vegetation NPP change, and references have been added. This equation should not be labeled only as MSAVI. It has been modified in line 197.
- Lines 182-183 These are methods, not results.
Answer: Thanks for comment. This part has been moved to methods part in section 2.3. See line 169-170.
- Table 2 How was accuracy calculated? Is it really model fit?
Answer: Accuracy result was obtained by model fitting. This section has been updated. See line 217-224.
- Lines 214-224 Refer to Figure 1 since readers might not be familiar with geographic locations in the study area.
Answer: This section has been updated and Figure 1 is revised as well. See line 125 and line 241-242.
- Line 224 What is meant by ‘plaque’?
Answer: ‘plaque’ has been replaced by ‘patche’. See line 248.
-Figure 3 needs a legend (what is the red line?) and detailed caption. If authors want to describe a trend, add a trend line to the chart.
Answer: Figure 3 has been updated. (The red line is the average grassland production for ten years.) See line 262.
- Tables 3 and 4 should acknowledge potentially large error in biomass estimates since best fit model has R-square = 0.72 (and accuracy 78%) that leaves 20%+ uncertainty in the results. Should be clearly spelled out.
Answer: Thanks. This section has been revised. See line 281-282.
- Figure 4 Legend is not clearly labeled. What is ‘coefficient’?
Answer: Thanks. Figure 4 has been updated. See line 288.
- Line 262 What units are ‘0 and 1’?
Answer: The unit here should be kg*ha-²*a-1, which has been updated. See line 292.
- Line 263 Instead of ‘green patches’ use what the green patches mean (more biomass increase?)
Answer: Green patches indicate areas where the growth of grassland production is more significant. This section has been clearly stated. See line 294-295.
- Line 276 Remove ‘s’ from ‘Discussions’
Answer: Thanks. The ‘S’ has been removed. See line 308.
- Line 278 There are many more than 40 SVIs: https://www.indexdatabase.de/db/i.php
Answer: Thanks. This section has been revised. See line 310-311.
- Line 325 What is meant by ‘obvious’? Significant?
Answer: It means Significant. ‘Obvious’ has been replaced by ‘significant’. See line 359.
- Line 342 Acknowledge potential error in the results. A lot of uncertainty. Also not clear how well model trained on one year of data really predicts other years.
Answer: Thank you for comments. We have added the potential error statement in the results. See line 281-282 and line 378-379. As for the predicts for years, sampling work can only be done in the current year, we cannot get the sampling data of previous years, so we used the ground data that had collected recent years to build the estimation model. In the future, we will further collect sampling data, pursue better models for grassland production retrieving through a longer time-series study.

Reviewer 2 Report
This is an interesting paper dealing with relevant topics.
In the attached pdf file of the annotated paper some changes, clarifications and integrations are suggested with the aim to improve the quality of the paper.
It is necessary to explain more accurately what kind of temperature and rainfall data were used and how they were used and treated.
Moreover, it is necessary to spend some lines to describe also the way you used the DEM (did you used elevation data or did you consider other data extracted from dem?).
It is not clear how climatic and elevation (?) data influenced the grassland production, thus the obtained results. For this reason further explanations concerning these elements are recommended.
Adding these missing elements this research could be considered complete.

Author Response
Comments and Suggestions for Authors
This is an interesting paper dealing with relevant topics.
In the attached pdf file of the annotated paper some changes, clarifications and integrations are suggested with the aim to improve the quality of the paper.
Answer: Thanks. We have revised the paper according to other comments marked in the PDF file.
-It is necessary to explain more accurately what kind of temperature and rainfall data were used and how they were used and treated.
Answer: Thanks for comments. The temperature and precipitation data have been clearly stated. See line 146-149 and line 169-170.
-Moreover, it is necessary to spend some lines to describe also the way you used the DEM (did you used elevation data or did you consider other data extracted from dem?).
Answer: Thanks for comments. This section has been revised. See line 150 and line 175-178. We extracted the DEM elevation data of sampling points for correlation analysis, but found that the correlation between DEM elevation and grassland production was not significant in the Mongolia Plateau where altitude was not very different, so it was not used for modeling.
-It is not clear how climatic and elevation (?) data influenced the grassland production, thus the obtained results. For this reason further explanations concerning these elements are recommended.
Answer: Thanks for comments. This section has been revised. See line 169-172 and line 175-182. The altitude in the study area was not very different, and the impact on grassland production was not significant, so it was not used for modeling.
Adding these missing elements this research could be considered complete.
Answer: Thanks for comments. We have added all these missing elements in the manuscript.

Round 2
Reviewer 1 Report
REVISION
Overview:
Still not addressed: methods are not clearly explained and the results are overstated given the low accuracy numbers being reported.
- The literature review seems quite brief - are there no more sources available?
Still seems quite short.
- Figure 1 needs a context map and more detailed caption.
New map still provides no context for readers outside the region.
- Line 118 Define the MODIS data products and provide a reference. I meant define what these MODIS products are- MODIS13Q1 and MODIS17A2H- Not MODIS products in general. Also should be MOD13Q1 and MOD17A2H Maybe use these pages: https://lpdaac.usgs.gov/dataset_discovery/modis/modis_products_table/mod13q1_v006
https://lpdaac.usgs.gov/dataset_discovery/modis/modis_products_table/mod17a2h_v006
- Line 224 What is meant by ‘plaque’? Should be ‘patch’ then?
Answer: ‘plaque’ has been replaced by ‘patche’. See line 248.
-Figure 3 needs a legend (what is the red line?) and detailed caption. If authors want to describe a trend, add a trend line to the chart. Still no legend indicating what either lines shows.
Answer: Figure 3 has been updated. (The red line is the average grassland production for ten years.) See line 262.
- Line 263 Instead of ‘green patches’ use what the green patches mean (more biomass increase?)
Answer: Green patches indicate areas where the growth of grassland production is more significant. This section has been clearly stated. See line 294-295. Line numbers end after line 286
Author Response
REVISION Overview: Still not addressed: methods are not clearly explained and the results are overstated given the low accuracy numbers being reported. - The literature review seems quite brief - are there no more sources available? Still seems quite short. Answer: Thanks for comments. We have added more references on biomass estimation in this section, see line 78-80, 86-89, and line 97-101. - Figure 1 needs a context map and more detailed caption. New map still provides no context for readers outside the region. Answer: Thanks for comment. Figure 1 has been updated, see line 135. - Line 118 Define the MODIS data products and provide a reference. I meant define what these MODIS products are- MODIS13Q1 and MODIS17A2H- Not MODIS products in general. Also should be MOD13Q1 and MOD17A2H Maybe use these pages: https://lpdaac.usgs.gov/dataset_discovery/modis/modis_products_table/mod13q1_v006 https://lpdaac.usgs.gov/dataset_discovery/modis/modis_products_table/mod17a2h_v006 Answer: Thanks for comment. We have defined MODIS13Q1 and MODIS17A2H in this section, see line 142-150. - Line 224 What is meant by ‘plaque’? Should be ‘patch’ then? Answer: Plaque means a small abnormal patch on or inside the body. Plaque is not suitable here, patch can better express the meaning of this sentence. We have replaced‘plaque’ with‘patch’ in the full paper, see line 264, 266 and line268. -Figure 3 needs a legend (what is the red line?) and detailed caption. If authors want to describe a trend, add a trend line to the chart. Still no legend indicating what either lines shows. Answer: Thanks. Figure 3 has been updated (Legend has been added.), see line 281. - Line 263 Instead of ‘green patches’ use what the green patches mean (more biomass increase?) Answer: Green patches indicate areas where the growth of grassland production is more significant. This section has been clearly stated. See line 294-295. Line numbers end after line 286 Answer: Line number has been revised, see line 312-313 for this part. Submission Date 27 February 2019 Date of this review 21 Mar 2019 21:59:13

Reviewer 2 Report
Thanks for answers and changes.
The authors have only partially improved their manuscript. Some information concerning meteorological data and dem data are now provided, but these data don’t appear in the text. In particular, the passage from data (temperature, rainfall and elevation data) to the correlation analysis, and thus to the definition of factors influencing grassland growth) is not explained enough. Moreover, in lines 177-182 it is not clear how data were processed in GIS (I mean before the specified “overlay”). We think that something more could be inserted (for example the rainfall interpolation type; the number of gauging stations considered for interpolation should be also specified).
Author Response
Comments and Suggestions for Authors
Thanks for answers and changes.
The authors have only partially improved their manuscript. Some information concerning meteorological data and dem data are now provided, but these data don’t appear in the text. In particular, the passage from data (temperature, rainfall and elevation data) to the correlation analysis, and thus to the definition of factors influencing grassland growth) is not explained enough. Moreover, in lines 177-182 it is not clear how data were processed in GIS (I mean before the specified “overlay”). We think that something more could be inserted (for example the rainfall interpolation type; the number of gauging stations considered for interpolation should be also specified).
Answer: Thanks for comments. We have added meteorological data and DEM elevation data information to the paper, and further explain the data and correlation analysis, see line 165-167 and line 231-234. The processing of meteorological data has also been added, see line 192-194.

Round 3
Reviewer 1 Report
REVISION 2
New comments are highlighted.
Overview: Still not addressed: methods are not clearly explained and the results are overstated given the low accuracy numbers being reported. -
This was not addressed from what I could see and newest version does not indicate changes in red text so changes were hard to track.
The literature review seems quite brief - are there no more sources available? Still seems quite short.
Answer: Thanks for comments. We have added more references on biomass estimation in this section, see line 78-80, 86-89, and line 97-101. -
Okay
Figure 1 needs a context map and more detailed caption. New map still provides no context for readers outside the region.
Answer: Thanks for comment. Figure 1 has been updated, see line 135. -
Okay
Line 118 Define the MODIS data products and provide a reference. I meant define what these MODIS products are- MODIS13Q1 and MODIS17A2H- Not MODIS products in general. Also should be MOD13Q1 and MOD17A2H Maybe use these pages: https://lpdaac.usgs.gov/dataset_discovery/modis/modis_products_table/mod13q1_v006 https://lpdaac.usgs.gov/dataset_discovery/modis/modis_products_table/mod17a2h_v006 Answer: Thanks for comment. We have defined MODIS13Q1 and MODIS17A2H in this section, see line 142-150. -
Okay, but product names are still wrong: MOD13Q1 and MOD17A2H (not MODIS13Q1 and MODIS17A2H)
Line 224 What is meant by ‘plaque’? Should be ‘patch’ then?
Answer: Plaque means a small abnormal patch on or inside the body. Plaque is not suitable here, patch can better express the meaning of this sentence. We have replaced‘plaque’ with‘patch’ in the full paper, see line 264, 266 and line268. -
Okay, but the word ‘patche’ is still in several places and this is not a word as far as I know.
Figure 3 needs a legend (what is the red line?) and detailed caption. If authors want to describe a trend, add a trend line to the chart. Still no legend indicating what either lines shows.
Answer: Thanks. Figure 3 has been updated (Legend has been added.), see line 281. -
Okay
Line 263 Instead of ‘green patches’ use what the green patches mean (more biomass increase?)
Answer: Green patches indicate areas where the growth of grassland production is more significant. This section has been clearly stated. See line 294-295.
Line 294 is Table 3 not text. I still see the language being used in Line 310 instead of what “green patches” are- please
Line numbers end after line 286 Answer: Line number has been revised, see line 312-313 for this part.
Okay
Author Response
New comments are highlighted. Overview: Still not addressed: methods are not clearly explained and the results are overstated given the low accuracy numbers being reported. - This was not addressed from what I could see and newest version does not indicate changes in red text so changes were hard to track. Answer: Thanks for comments. We have added a further description in method section, results section, and conclusions section using red color texts. In 2.3, we have added a further description of the method in the previous version, see line 190-196. In the section 3 results, the potential uncertainty of the best fit model has been further explained, see line 237-241, 243-248. The conclusions have been updated to describe the uncertainty in model, see line 392-393, 405-406. The literature review seems quite brief - are there no more sources available? Still seems quite short. Answer: Thanks for comments. We have added more references on biomass estimation in this section, see line 78-80, 86-89, and line 97-101. - Okay Answer: Thanks. Figure 1 needs a context map and more detailed caption. New map still provides no context for readers outside the region. Answer: Thanks for comment. Figure 1 has been updated, see line 135. - Okay Answer: Thanks. Line 118 Define the MODIS data products and provide a reference. I meant define what these MODIS products are- MODIS13Q1 and MODIS17A2H- Not MODIS products in general. Also should be MOD13Q1 and MOD17A2H Maybe use these pages: https://lpdaac.usgs.gov/dataset_discovery/modis/modis_products_table/mod13q1_v006 https://lpdaac.usgs.gov/dataset_discovery/modis/modis_products_table/mod17a2h_v006 Answer: Thanks for comment. We have defined MODIS13Q1 and MODIS17A2H in this section, see line 142-150. - Okay, but product names are still wrong: MOD13Q1 and MOD17A2H (not MODIS13Q1 and MODIS17A2H) Answer: Thanks. MODIS product names have been updated, see line 137,141,143,145 and 147. Line 224 What is meant by ‘plaque’? Should be ‘patch’ then? Answer: Plaque means a small abnormal patch on or inside the body. Plaque is not suitable here, patch can better express the meaning of this sentence. We have replaced‘plaque’ with‘patch’ in the full paper, see line 264, 266 and line268. - Okay, but the word ‘patche’ is still in several places and this is not a word as far as I know. Answer: Thanks. ‘Patche’ should be written as ‘patches’. We have revised it, see line 268,270 and 272. Figure 3 needs a legend (what is the red line?) and detailed caption. If authors want to describe a trend, add a trend line to the chart. Still no legend indicating what either lines shows. Answer: Thanks. Figure 3 has been updated (Legend has been added.), see line 281. - Okay Answer: Thanks. Line 263 Instead of ‘green patches’ use what the green patches mean (more biomass increase?) Answer: Green patches indicate areas where the growth of grassland production is more significant. This section has been clearly stated. See line 294-295. Line 294 is Table 3 not text. I still see the language being used in Line 310 instead of what “green patches” are- please Answer: The line number 294 is wrong here. Green patches indicate areas where grassland production increased more significant, see line 315-317. Line numbers end after line 286 Answer: Line number has been revised, see line 312-313 for this part. Okay Answer: Thanks. Submission Date 27 February 2019 Date of this review 31 Mar 2019 19:19:55